# CASF-Net: Underwater Image Enhancement with Color Correction and Spatial Fusion

**DOI:** 10.3390/s25082574

**Published:** 2025-04-18

**Authors:** Kai Chen, Zhenhao Li, Fanting Zhou, Zhibin Yu

**Affiliations:** 1Key Laboratory of Ocean Observation and Information of Hainan Province, Sanya Oceanographic Institution, Ocean University of China, Sanya 572000, China; ck03240217@163.com (K.C.); lzh02030300@163.com (Z.L.); zhouftft@163.com (F.Z.); 2Faculty of Information Science and Engineering, Ocean University of China, Qingdao 266100, China

**Keywords:** underwater image enhancement, channel adaptive factor, multi-scale fusion, spatial information fusion

## Abstract

With the exploration and exploitation of marine resources, underwater images, which serve as crucial carriers of underwater information, significantly influence the advancement of related fields. Despite dozens of underwater image enhancement (UIE) methods being proposed, the impacts of insufficient contrast and distortion of surface texture during UIE are currently underappreciated. To address these challenges, we propose a novel UIE method, channel-adaptive and spatial-fusion Net (CASF-Net), which uses a network channel-adaptive correction module (CACM) to enhance feature extraction and color correction to solve the problem of insufficient contrast. In addition, the CASF-Net utilizes a spatial multi-scale fusion module (SMFM) to solve the surface texture distortion problem and effectively improve underwater image saturation. Furthermore, we propose a Large-scale High-resolution Underwater Image Enhancement Dataset (LHUI), which contains 13,080 pairs of high-resolution images with sufficient diversity for efficient UIE training. Experimental results show that the proposed network design performs well in the UIE task compared with existing methods.

## 1. Introduction

Underwater imaging technology plays a critical role in the exploration of ocean resources [1]. Due to the unique characteristics of the underwater environment—such as light attenuation, scattering, and uneven illumination—underwater images often suffer from blurriness, low contrast, and color distortion [2]. These issues not only affect the visual quality of the images but also severely limit their applications in fields like ocean exploration, underwater target detection, and underwater robotic operations. Therefore, developing underwater image enhancement (UIE) technology [3] has become urgent, with the goal of restoring high-quality images by removing degradations like noise, blurriness, and color deviations.

The degradation of underwater images mainly arises from light scattering and absorption caused by suspended particles in water, as well as the varying absorption capacities of water for different wavelengths [4,5]. Water molecules tend to absorb red light, while suspended particles primarily absorb green light, causing underwater images to appear blue-green. Additionally, the imaging depth influences color deviation, giving images a hazy appearance with low-contrast colors. Consequently, UIE technology has emerged. As ocean resource development deepens, the demand for higher-quality underwater images is increasing, further driving the advancement and innovation of UIE technology.

To address severe underwater image degradation, early studies attempted non-physical model-based methods [6,7,8], which directly process pixel values. These methods are simple, fast, and easy to implement, but they may lead to over-enhancement, oversaturation, and color distortion in the resulting images. Subsequently, physics-based UIE methods [9,10,11] were proposed. These methods have shown promising performance in enhancing images from specific underwater scenes, but due to their inherent mechanisms, they lack flexibility and struggle to handle diverse underwater scenarios.

In recent years, given the powerful potential of neural networks, researchers have proposed GAN and CNN-based approaches [12,13,14,15,16,17] to enhance underwater images. Compared to the traditional enhancement methods mentioned above, GAN and CNN methods demonstrate much stronger performance. However, these methods often treat different image channels uniformly, overlooking the inter-channel differences in feature extraction. Figure 1 shows the results of our UIE method and some comparison UIE methods, and the main contributions of this paper can be summarized as follows:We introduce the network channel adaptive correction module (CACM), which introduces an adaptive factor to solve the problem of insufficient contrast and effectively improve the contrast of underwater images.We introduced the spatial multi-scale fusion module (SMFM) to process spatial information of different scales to solve the problem of surface texture distortion and effectively improve underwater image saturation.We propose a novel UIE method, CASF-Net, and conduct extensive comparative experiments on the LHUI underwater dataset. The experimental results show that our method outperforms other methods in both qualitative and quantitative aspects.

**Figure 1 sensors-25-02574-f001:**
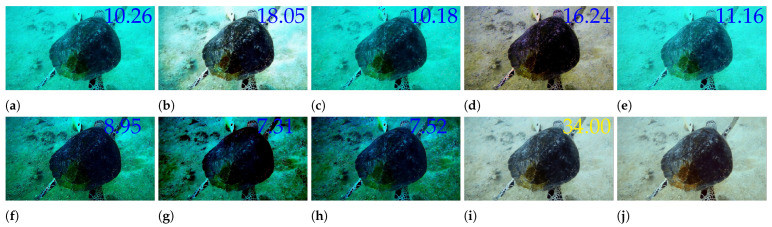
Compared with the existing UIE methods, the image produced by our CASF-Net has the highest PSNR score and best visual quality. The value 34.00 represents the PSNR values, and the yellow font indicates the highest values. (**a**) Raw image. (**b**) CLUIE [18]. (**c**) DCP [10]. (**d**) FspiralGAN [15]. (**e**) GC [19]. (**f**) GDCP [20]. (**g**) MSCNN [16]. (**h**) UDCP [9]. (**i**) Ours. (**j**) Ground truth.

## 2. Related Work

In the domains of ocean resource exploration and underwater robotic operations, UIE technology plays a vital role. Recent years have witnessed notable advancements in research and development within this field [21]. Presently, UIE techniques are primarily categorized into two groups: traditional enhancement methods and those based on deep learning. Furthermore, the creation and utilization of datasets are critical in UIE research. With the progress in underwater imaging technology, numerous underwater image datasets have been established in recent years, providing valuable resources for data-driven deep learning approaches and further propelling the advancement of UIE technology.

### 2.1. Traditional Enhancement Algorithms

Traditional underwater enhancement methods can be classified into non-physical model-based methods and physical model-based methods. Non-physical model-based methods focus on enhancing global contrast by adjusting the histogram distribution of an image, achieving a more even pixel value distribution to improve the image’s visual quality. These methods do not rely on complex underwater optical models but instead operate directly on pixel values to obtain clearer underwater images. [6] combined histogram equalization (HE) with white balance, first applying white balance to correct color casts in blurred images, followed by histogram equalization. This approach leverages the strengths of both techniques, achieving excellent results in color restoration and contrast enhancement. Land et al. [7] introduced the concept of “color constancy” in the human visual system, emphasizing the importance of ambient light in color perception and laying the foundation for the multi-scale Retinex algorithm. Based on the Retinex principle, an enhancement method [8] was developed for single-frame underwater images, effectively improving image brightness and color performance. These non-physical model-based methods are simple, easy to use, and require low computational power, making them particularly suitable for resource-constrained environments. However, since these methods operate directly on the pixels of the image, they may introduce detail loss during the enhancement process, resulting in insufficient accuracy in the local information of the image.

Physics-based UIE methods focus on restoring the true characteristics of an image by simulating the physical phenomena associated with underwater imaging. These approaches treat UIE as an inversion problem, aiming to reverse the degradation caused by the underwater imaging process in order to reconstruct high-quality images. This approach requires estimating parameters of the underwater imaging model, such as background light and transmission maps, to reconstruct the image through inverse degradation processes. For example, method [9] introduced the UDCP algorithm for UIE, inspired by methods [10]. By analyzing the underwater environment, UDCP improves the traditional DCP algorithm, generating clearer underwater images. Wang et al. [11] proposed a new adaptive attenuation-curve prior model based on the adaptive attenuation-curve prior, which adjusts the attenuation curve according to the specific underwater environment to more accurately estimate light attenuation. Physics-based UIE methods restore images by simulating the physical processes of underwater imaging, providing relatively realistic restoration results. However, these methods have limitations, such as difficulty in accurately estimating model parameters affected by multiple factors in real applications. The complex calculations involved can lead to slower processing speeds, and these methods are highly dependent on specific underwater scenes, requiring parameter adjustments for different environments.

### 2.2. Deep-Learning-Based Methods

As underwater imaging technology has advanced, more datasets of underwater scenes are being published, creating favorable conditions for data-driven deep learning methods. Generative adversarial networks (GANs) and convolutional neural networks (CNNs) have shown tremendous potential in the field of UIE. Li et al. [12] first introduced WaterGAN, a method that converts non-underwater scenes into underwater scenes to generate training data. By training models on these generated underwater images along with real underwater images, the model can enhance murky underwater images into clearer ones. Fabbri et al. [13] proposed the UGAN network architecture, which preserves spatial information generated by the encoder, enabling it to learn information from input images without relying on the final output feature map, thus producing high-quality images. Li et al. [14] introduced a novel underwater image color correction method based on weakly supervised color transfer. Han et al. proposed the FspiralGAN architecture [15], which uses an “equal channel strategy” to improve model speed while maintaining the quality of generated images. Due to the capabilities of convolutional neural networks, numerous CNN-based methods have been developed. Among them, Ren et al. [16] proposed a multi-scale convolutional neural network (MSCNN), which extracts features at different scales through parallel multi-scale convolution layers, effectively improving detail representation in underwater images, allowing the model to handle complex details across diverse underwater environments. Additionally, Fu et al. [17] proposed PUIE, a perception-driven image enhancement method that designs a perceptual loss function, making the enhanced images visually more aligned with human perception. The PUIE method emphasizes color fidelity and contrast enhancement, adaptively adjusting image details in various underwater scenes and improving color distortion and turbidity issues to make the enhanced images appear more natural and realistic. However, existing image enhancement methods do not fully consider the problem of attenuation differences among underwater color channels and still suffer from insufficient contrast, surface texture distortion, and insufficient saturation. To address this, we propose a new method, CASF-Net, which includes two modules: CACM and SMFM, and demonstrates strong performance on the LHUI dataset.

### 2.3. Underwater Image Datasets

Capturing underwater images is inherently challenging due to the complexities of the underwater environment, making it difficult to obtain true, undistorted scenes. In the early phases of computer vision and underwater image processing, researchers created underwater image datasets that were often limited in quantity, focused on single scenes, and lacked reference features. However, as technology advanced, it became easier to acquire underwater images, and deep learning techniques demonstrated remarkable effectiveness in image processing. Researchers subsequently developed UIE methods to address the blurriness of underwater images and carefully selected images that closely resembled realistic underwater scenes, resulting in the creation of numerous datasets that have significantly contributed to the advancement of UIE technology. For instance, ref. [22] introduced the EUVP dataset, ref. [23] presented the UIEB dataset, and [24] developed the LSUI dataset. However, although many datasets have been proposed, they still face the issue of limited quantity. To address this problem, we propose a new dataset, LHUI.

## 3. Dataset

**Data Collection.** The UVEB [25] dataset is a large-scale video collection that includes a significant number of underwater video pairs. However, these video pairs, which include many similar frames, are less effective and time-costly for UIE training. To tackle this challenge, we implemented a frame extraction process to create a more lightweight dataset tailored for image enhancement tasks, referred to as LHUI. Specifically, 10 frames were extracted from each video: for those with more than 50 frames, one frame was taken every five frames; for videos with fewer than 50 frames, the first 10 frames were selected. The LHUI dataset is comprised of two primary components. The first part features underwater videos captured using FIFISH V6 and FIFISH V-EVO cameras, both offering 4 K resolution. These videos were recorded in various marine locations and ports throughout China, representing 55% of the dataset. The second component consists of images obtained from the internet, contributed by underwater photographers, which make up the remaining 45% of the dataset. Overall, the dataset contains 13,080 image pairs, and all images are intended strictly for academic use. To ensure the diversity and applicability of the dataset, we prioritized selecting authentic underwater images with rich water scenes, diverse water types, varying lighting conditions, and high resolutions. These images were further processed to generate clear reference images for image enhancement tasks.

**Reference Image Generation.** The selection of reference images is similar to [25]. We first used 20 existing optimal UIE methods, including [6,8,9,10,15,16,17,18,19,20,26,27,28,29,30,31,32,33,34,35], to process the collected underwater images, creating a set of 20×13,080 images. The entire process of selecting the best-enhanced images was conducted under the guidance of ITU-R BT500-13 [36], with 15 observers. All observers used the same experimental equipment, a Redmi-27H 4 K monitor, to perform image quality assessments, resulting in the optimal reference dataset, which serves as the ground truth (GT).

**Diversity Analysis.** LHUI encompasses a wide range of underwater scenes, including coastal areas, open seas, rivers, lakes, ports, aquariums, and swimming pools. Figure 2a illustrates the scene diversity within the LHUI samples, which mainly includes six types of underwater scenes: blue, green, yellow, white, low-light, and other color deviations. Other color refers to some challenging underwater degraded images that have issues with non-uniform spatial area attenuation, where a single image contains multiple color degradations. Among these, blue scenes have the highest proportion, accounting for 39.2%, followed by green scenes at 38.7%. Although scenes with yellow, white, and low-light conditions are less common, they still make up 22.2% of the dataset. Figure 2b shows the resolution distribution of the LHUI dataset, with the majority of images concentrated in the 2 K to 4 K range, comprising 65.6% of the dataset, followed by the 720 p to 1080 p range, making up 25.1%. Figure 3 compares the resolution distributions of the LHUI, LSUI [24], and UIEB [23] datasets. The LHUI dataset contains a total of 13,080 image pairs, of which 8580 pairs have resolutions above 2 K. In contrast, the LSUI [24] dataset, with a total of 4279 pairs, primarily features resolutions between 0 and 360 p. The UIEB [23] dataset has 890 image pairs, with resolutions mainly distributed between 360 p to 720 p and 720 p to 1080 p. To the best of the authors’ knowledge, LHUI is currently the largest real-world underwater image dataset, offering a large-scale, high-quality set of reference images that can further drive advancements in UIE methods.

## 4. Network Architecture

### 4.1. Overall Pipline

As shown in Figure 4, the proposed CASF-Net is essentially a single-branch network primarily composed of two modules: CACM and SMFM. First, $X_{i}$ represents the input degraded frame image, which undergoes preliminary feature extraction through a 3×3 convolution. Then, feature $L_{i}$ is obtained by passing it through 10 residual blocks. Next, $L_{i}$ is processed by the CACM to obtain the enhanced feature $M_{i}$. Subsequently, $M_{i}$ passes through 15 SMFM blocks, effectively utilizing spatial and multi-scale receptive field information along with feature fusion techniques to obtain the feature $R_{i}$, which can be expressed as:(1)$$R_{i}=SMFM\left(CACM\left(L_{i}\right)\right)$$

Finally, a 3×3 convolution is applied to obtain the output image $Y_{i}$. The proposed CACM module enhances the feature extraction and color correction functions to solve the problem of insufficient contrast, while the SMFM module uses spatial information to capture multi-scale receptive fields for feature fusion to solve the problem of distortion of surface texture and effectively improve the saturation of underwater images.

### 4.2. CACM

The RGB three-channel information of images has consistently garnered attention. Inspired by DCP [10] and UDCP [9], we combine channel information with deep learning algorithms to generate a network channel-adaptive factor, forming our CACM.

Specifically, the input feature $L_{i}$ first passes through a 2×2 convolution and an adaptive average pooling layer to extract more complex features. Then, it is processed through a 1×1 convolution, followed by the sigmoid activation function to generate the channel-adaptive factor $F_{i}$, which can be represented as:(2)$$F_{i}=Sigmod\left(Conv_{1}\left(AdaptiveAvgPool\left(Conv_{2}\left(L_{i}\right)\right)\right)\right)$$

The input feature $L_{i}$ is corrected to obtain $F_{i}^{\prime }$, which is then connected with Li via a residual connection to produce the output $M_{i}$. $F_{i}$ can be represented as:(3)$$M_{i}=L_{i}\cdot F_{i}+L_{i}$$

CACM performs adaptive modulation of degraded features by introducing network channel-adaptive factors for enhanced feature extraction and color correction to improve image contrast.

### 4.3. SMFM

Recently, in the field of computer vision [37,38,39,40], spatial and multi-scale receptive field information has become a popular topic. Inspired by [41,42,43,44], we utilize channel separation and a multi-scale mechanism, combined with fusion techniques, to form our SMFM.

Specifically, the input feature $M_{i}$ is processed through a 3×3 convolution followed by a ReLU activation function. We then split it evenly along the channel dimension into two parts, $Z_{i}^{\left(1\right)}$ and $Z_{i}^{\left(2\right)}$. After standard convolution and activation, $Z_{i}^{\left(1\right)}$ produces $E_{i}^{\left(1\right)}$. For $Z_{i}^{\left(2\right)}$, we apply strided convolution to reduce the feature resolution to half the original, then perform 2x upsampling to restore it to the original size, resulting in feature $E_{i}^{\left(2\right)}$. The two features from the upper and lower branches are concatenated and finally connected to the residual of feature $M_{i}$, producing the output $R_{i}$, which can be represented as:(4)$$R_{i}=E_{i}+M_{i}$$

SMFM adopts a channel separation technique to fully utilize the spatial and multi-scale sensory field information and utilizes residual connections to improve learning performance and efficiency, which solves the distortion problem of surface texture and effectively improves the saturation of underwater images.

### 4.4. Loss Function

In the training process, we use a dual-domain L1 loss function in the spatial and frequency domains, respectively [45,46]. Since the network modules mainly focus on the processing of pixel-level and inter-channel information of blurred images, it leads to insufficient attention to frequency domain information by the model, thereby losing some important details during the image enhancement process. The dual-domain loss function can consider both spatial domain and frequency domain information during the training process, ensuring that the model does not overly ignore frequency domain features during training, thereby improving the quality of the model in enhancing images. For each output/target image pair with the same resolution, loss functions are given by:(5)$$L=L_{1}\left(y^{\prime },y\right)+\lambda L_{fft}\left(F\left(y^{\prime }\right),F\left(y\right)\right)$$
where $L_{1}$ represents the L1 loss, and $L_{fft}$ denotes the frequency domain loss; $y^{\prime }$ and *y* denote the output and GT images, respectively; *F* represents fast Fourier transform; and λ is empirically set to 0.1 for balancing dual-domain training.

## 5. Experiments

### 5.1. Settings

**Datasets.** LHUI contains 12,080 pairs of training images and 1000 pairs of test images, featuring six types of underwater degradation.**Comparison methods.** We compared CASF-Net with 12 UIE methods to verify our performance advantage.**Evaluation metrics.** For the test dataset with reference images, we conducted a full-reference evaluation using the PSNR [47] and SSIM [48] metrics. These two metrics reflect the degree of similarity to the reference, where a higher PSNR [47] value indicates closer image content, and a higher SSIM [48] value reflects a more similar structure and texture.

PSNR is an objective measure used to assess the difference between two images. It reflects the ratio between the signal (the original image) and the noise (the distorted part). The formula for its calculation is:(6)$$\mathrm{PSNR}=10\cdot \log_{10}\left(\frac{{\mathrm{MAX}}^{2}}{\mathrm{MSE}}\right)$$MAX is the maximum possible pixel value in an image.

SSIM is an indicator tric measure of the similarity in brightness, contrast, and structure of two images. The core idea of SSIM is to view images as a collection of brightness, contrast, and structure, and to evaluate the overall similarity by comparing the similarity of these three aspects. The calculation formula is as follows:(7)$$\mathrm{SSIM}\left(x,y\right)=\frac{\left(2\mu_{x}\mu_{y}+C_{1}\right)\left(2\sigma_{xy}+C_{2}\right)}{\left(\mu_{x}^{2}+\mu_{y}^{2}+C_{1}\right)\left(\sigma_{x}^{2}+\sigma_{y}^{2}+C_{2}\right)}$$ In which $\mu_{x}$ and $\mu_{y}$ are the mean values of images *x* and *y* within a local window, representing the brightness level of the images. $\sigma_{x}^{2}$ and $\sigma_{y}^{2}$ are the variances of images *x* and *y* within the local window, representing the contrast of the images. $\sigma_{xy}$ is the covariance between images *x* and *y* within the local window, representing the structural similarity of the images. $C_{1}={\left(K_{1}L\right)}^{2}$, $C_{2}={\left(K_{2}L\right)}^{2}$, where $K_{1}=0.01$, $K_{2}=0.03$, and *L* is the dynamic range of the pixel values.

**Implementation details.** We implemented our method using PyTorch and trained it on an NVIDIA Tesla A40 GPU. The network optimization was performed using the ADAM [49] optimizer. The initial learning rate was set to $9\times 10^{-5}$. The total number of iterations was 2 K. The batch size was 128, and the block size of the input image was 128 × 128.**Algorithm introduction.** We briefly introduce the contrast methods used, as shown in Table 1.

### 5.2. Comparisons with State-of-the-Art Methods

**Quantitative comparison.** Table 2 shows the results of quantitative experiments for our proposed network, as well as the results of current state-of-the-art UIE methods. The experiments were conducted on our proposed LHUI dataset. Our proposed network achieved good performance among all methods, with the best results in terms of PSNR [47], SSIM [48], and MSE.

**Table 1 sensors-25-02574-t001:** A brief introduction to the underwater image enhancement algorithm used.

Methods	Reference	Introduction
LiteEnhanceNet	[50]	Lightweight CNN Network Model Based on Depthwise Separable Convolution
LANet	[26]	Image Enhancement Algorithm Using Multi-Scale Spatial Information and Parallel Attention Mechanism
CLUIE	[18]	Underwater image enhancement algorithms with multiple reference learning
GC	[19]	Based on the basic knowledge of human vision
MSCNN	[16]	Image Dehazing Method Based on Multi-Scale Deep Neural Networks
DCP	[10]	Image dehazing method using the dark channel prior
FspiralGAN	[15]	Adopting a GAN network model with an equal-channel design
CLAHE	[27]	Adaptive Histogram Equalization Enhancement Method
MetaUE	[33]	Model-based Underwater Image Enhancement Algorithm
GDCP	[20]	Universal Image Restoration Algorithm Using the Dark Channel Prior
UDCP	[9]	A method for estimating transmission in underwater environments has been proposed, along with a corresponding underwater image enhancement algorithm
PUIE	[17]	Underwater Image Enhancement Method Based on Probabilistic Networks

**Qualitative comparison.** Figure 5 presents a comparison of our image enhancement results with state-of-the-art methods on the LHUI dataset. It is evident that our method produces images with the best sharpness, contrast, and detail restoration, closely matching the GT. When handling underwater images with different color domains, many methods perform well only on degraded images within a specific color domain, revealing a lack of generalization ability. Figure 6 compares the three-channel color histograms of the images with the reference image. It is evident that our results are closest to the RGB color space distribution. In contrast, our method achieves the best enhancement results across images from various color domains, fully demonstrating the excellent generalization capability of the proposed model.

**Table 2 sensors-25-02574-t002:** Quantitative comparison with state-of-the-art methods. ↑ indicates that higher values are more desirable, and ↓ indicates that lower values are more desirable. Top 1st, 2nd results are marked in red and blue, respectively.

Methods	PSNR (dB) ↑	SSIM ↑	MSE $\times \;10^{2}↓$
LiteEnhanceNet [50]	24.86	0.9175	0.8128
LANet [26]	21.36	0.9142	1.1893
CLUIE [18]	18.92	0.8877	1.8170
GC [19]	15.65	0.8421	3.8121
MSCNN [16]	13.41	0.7568	5.6020
DCP [10]	12.67	0.7849	6.3114
FspiralGAN [15]	21.14	0.8507	1.8449
CLAHE [27]	19.43	0.9086	1.5046
MetaUE [33]	15.69	0.7937	3.0617
GDCP [20]	13.76	0.8209	5.2951
UDCP [9]	10.55	0.5591	10.1081
PUIE [17]	23.39	0.9302	0.7860
Ours	26.41	0.9401	0.6881

### 5.3. Ablation Studies

In our study, we designed and conducted a series of ablation experiments to deeply explore the practical effectiveness of the core components of our proposed network. As shown in Table 3, our experiments focused on evaluating two key factors: CACM and SMFM. We first experimented with the baseline model, achieving a PSNR of 23.84 and an SSIM of 0.9226. Next, we added the proposed CACM module to the baseline model, which increased the PSNR to 26.17 and the SSIM to 0.9325, clearly demonstrating the effectiveness of the CACM module. Further, we added the SMFM module to the model, resulting in the best performance, with a PSNR of 26.41 and an SSIM of 0.9401.

As shown in Figure 7, the enhancement result of the Full model has the highest PSNR and SSIM, with the best visual effect. The result of BL has more obvious blurring and haze, the result of BL + CACM has enhanced contrast and reduced blurring compared with BL, and the channel-adaptive factor is conducive to adaptive adjustment of degraded features, which is convenient to achieve the solution of the problem of low contrast. Full Model joins the SMFM module, the texture and color saturation of coral are more obvious, the sense of hierarchy is richer, close to GT, and the use of spatial information capture multi-scale feeling field for feature fusion is conducive to solving the problem of surface texture distortion, and effectively improves the saturation of the underwater image.

This series of experiments not only validated the effectiveness of the proposed modules but also highlighted the superiority of our proposed network.

**Figure 7 sensors-25-02574-f007:**
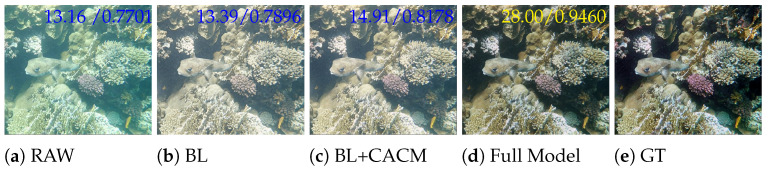
Visual comparison of the ablation study sampled from the LHUI dataset. BL represents the baseline, Full Model includes baseline + CACM + SMFM, 28.00/0.9460 represents the PSNR/SSIM values, and the yellow font indicates the highest values.

## 6. Conclusions

In this work, we created a new large-scale, high-resolution dataset, LHUI, containing 13,080 real-world underwater images. Compared to existing underwater datasets, LHUI features a greater variety of underwater scenes, types of underwater degradation, and high-resolution images, with corresponding clear images provided as references. Additionally, we propose a new image enhancement method, CASF-Net. This network includes two main modules: the CACM and SMFM. By combining channel information with deep learning algorithms and fully utilizing spatial and multi-scale receptive field information, CASF-Net achieves state-of-the-art UIE performance. Extensive experiments have verified that the network is able to solve the problems of insufficient contrast and surface texture distortion, and it is beneficial in improving the saturation of underwater images. However, the dataset may not comprehensively cover all possible underwater environments and conditions, such as low-light deep-sea scenarios. Therefore, we plan to incorporate other general enhancement techniques, such as low-light enhancement [35], in future work. In this process, we may further improve our CACM and SMFM modules, or introduce new modules to achieve better enhancement effects.

## Figures and Tables

**Figure 2 sensors-25-02574-f002:**
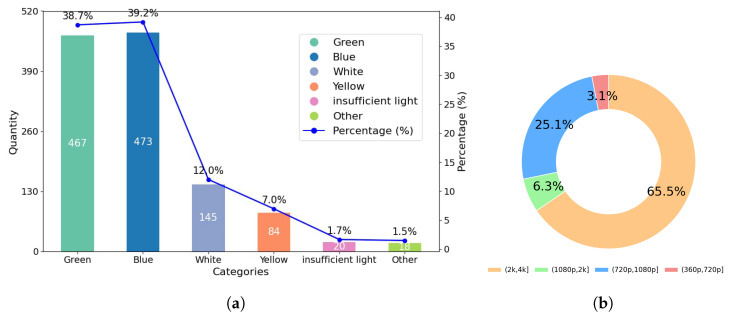
LHUI data analysis. (**a**) The diversity of LHUI samples includes six types of color degradation and their respective proportions. (**b**) The resolution distribution of LHUI.

**Figure 3 sensors-25-02574-f003:**
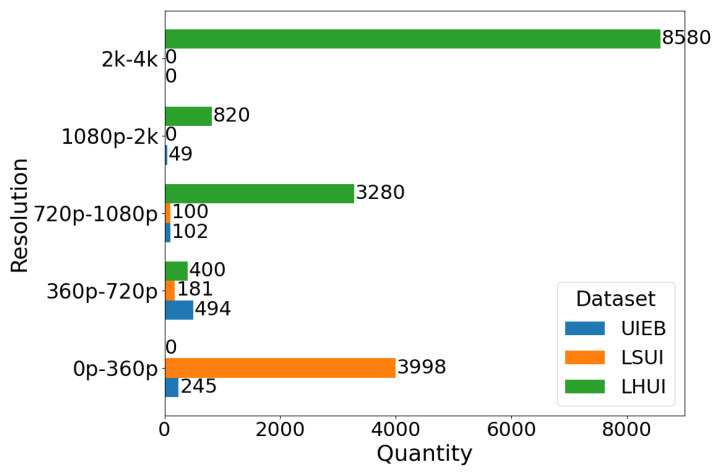
Comparison of quantity and resolution information among LHUI, UIEB [23], and LSUI [24].

**Figure 4 sensors-25-02574-f004:**
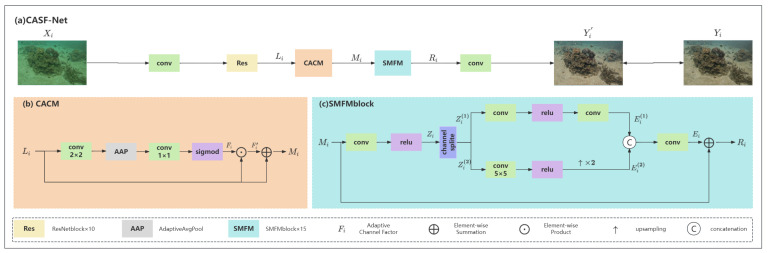
Overview of the proposed CASF-Net architecture. CACM and SMFM represent the network Channel-Adaptive Correction Module and the Spatial Multi-scale Fusion Module, respectively. Res, AAP, and $F_{i}$ represent the ResNet residual structure, adaptive average pooling, and network channel-adaptive factors, respectively.

**Figure 5 sensors-25-02574-f005:**
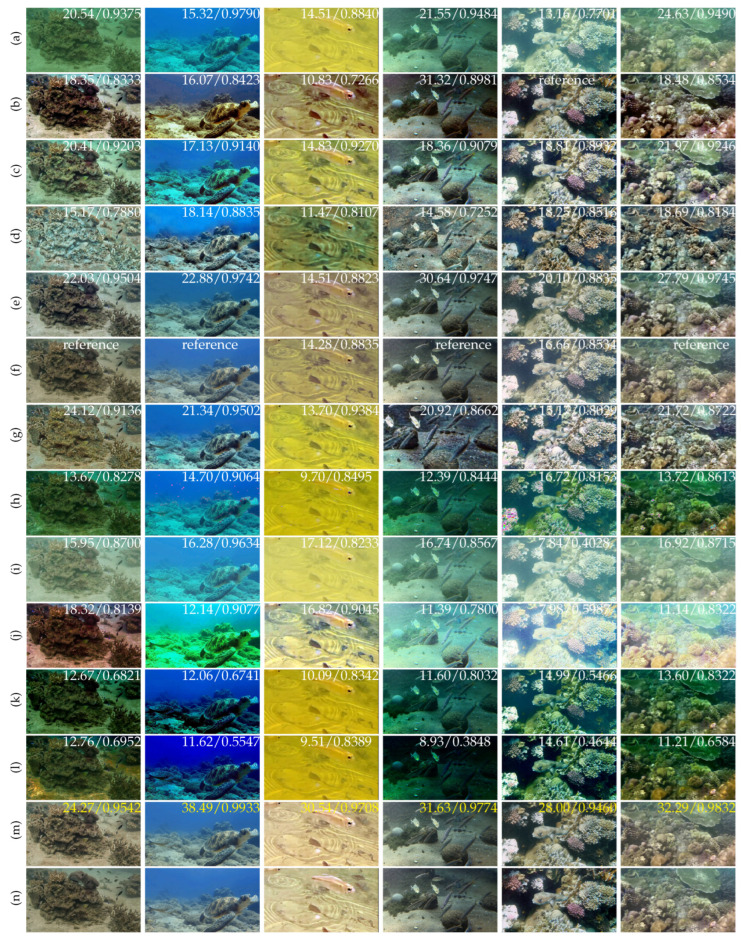
Visual comparisons with state-of-the-art methods on real underwater scenes. 32.29/0.9832 represents the PSNR/SSIM values, and the yellow font indicates the highest values. reference means that the image is the source of GTs. (**a**) RAW images. (**b**) Fspiral-GAN [15]. (**c**) CLAHE [27]. (**d**) MetaUE [33]. (**e**) PUIE [17]. (**f**) LANet [26]. (**g**) CLUIE [18]. (**h**) DCP [10]. (**i**) GC [19]. (**j**) GDCP [20]. (**k**) MSCNN [16]. (**l**) UDCP [9]. (**m**) Ours. (**n**) Ground Truth.

**Figure 6 sensors-25-02574-f006:**
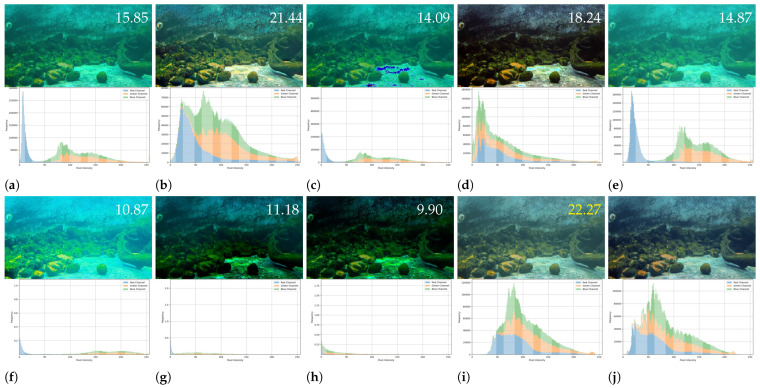
Comparison chart of three-channel color histograms. 22.27 represents the PSNR values, and the yellow font indicates the highest values. (**a**) Raw image. (**b**) CLUIE [18]. (**c**) DCP [10]. (**d**) FspiralGAN [15]. (**e**) GC [19]. (**f**) GDCP [20]. (**g**) MSCNN [16]. (**h**) UDCP [9]. (**i**) Ours. (**j**) Ground truth.

**Table 3 sensors-25-02574-t003:** Ablation studies. ↑ indicates that higher values are more desirable and ↓ indicates that lower values are more desirable. Top 1st results are marked in red.

	(a)	(b)	(c)
baseline	✓	✓	✓
CACM		✓	✓
SMFM			✓
PSNR (dB) ↑	23.84	26.17	26.41
SSIM ↑	0.9226	0.9325	0.9401
MSE $\times \;10^{-2}↓$	0.9525	0.7132	0.6881

## Data Availability

The dataset will be released on https://github.com/yzbouc/LHUI accessed on 15 April 2025.

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
