# Peer review of "CASF-Net: Underwater Image Enhancement with Color Correction and Spatial Fusion"

_sensors, 2025, doi:10.3390/s25082574_

Round 1
Reviewer 1 Report
Comments and Suggestions for Authors
The paper is very interesting and a good read. I would suggest to include the items below to further improve the quality. It should not been an issue to do so as the paper is rather short.
- The content of line 45 to 54 is repeated in line 58 to 66. The repeated elements should be removed. The use of dot points is preferable for readability reasons.
- Some repeat of "underwater image enhancement" even that UHI has been introduced previously (e.g. in line 96)
- Text in figures 1b and 3 is too small.
- In the experiment section, it is suggest to create a table with all the other UIE used, clearly stating their name, reference paper and a few words about the methods used.
- The evaluation metrics must be explained in more detail. How are they calculated? Why is it relevant?
- For the evaluation, is it possible to use some "true" reference comparison. Eg is it possible to take images of various structures in-air to compare the results to the in-air picture.
- The method of Adams (https://isprs-annals.copernicus.org/articles/X-4-2024/7/2024/isprs-annals-X-4-2024-7-2024.pdf) is not included in the evaluation section.
- How are the parameters trained for the other methods? Was the same setting used for all images? How were the parameters trained?
- I suggest that Table 1 and Figure 5 show the results of all tested methods.
Reviewer 2 Report
Comments and Suggestions for Authors
Please find the attached PDF.
